# IL-10 Producing B Cells Protect against LPS-Induced Murine Preterm Birth by Promoting PD1- and ICOS-Expressing T Cells

**DOI:** 10.3390/cells11172690

**Published:** 2022-08-29

**Authors:** Mandy Busse, Ana Claudia Zenclussen

**Affiliations:** 1Experimental Obstetrics and Gynecology, Medical Faculty, Otto-von-Guericke University, 39108 Magdeburg, Germany; 2Department of Environmental Immunology, Helmholtz Centre for Environmental Research-UFZ, 04318 Leipzig, Germany; 3Saxonian Incubator for Translation Research, Leipzig University, 04103 Leipzig, Germany

**Keywords:** B cells, IL-10, B10 cells, regulatory B cells, programmed cell death protein-1 (PD-), inducible costimulator (ICOS), preterm birth

## Abstract

B cells and in particular IL-10-secreting B cells emerge as important players in immune balance during pregnancy. We have recently revealed that CD19-deficient (CD19^−/−^), B cell-specific IL-10-deficient (BIL-10^−/−^) and B cell-deficient µMT pregnant mice are highly susceptible to LPS-induced preterm birth (PTB). We aimed to analyze the ability of IL-10-secreting cells to protect from PTB and the underlying mechanisms. Wild type (WT), CD19^−/−^, BIL-10^−/−^ and µMT mice were treated with LPS at gd16 and the cellular immune response was investigated 24 h later. LPS-treated BIL-10^−/−^ dams showed a more pronounced PTB phenotype compared to WT, CD19^−/−^ and µMT females, and increased inflammatory and reduced anti-inflammatory mediator concentrations in the peritoneal cavity and serum. CD19^−/−^, BIL-10^−/−^ and µMT mice displayed altered immune cell population frequencies in the blood and uterus with lower numbers of IL-10-secreting B cells and T cells. BIL-10^−/−^ mothers presented decreased frequencies of uterine CD4+CD25+Foxp3+ Treg cells. Co-stimulatory molecules are critical for feto-maternal tolerance and IL-10 secretion. We found dysregulated PD-1 expression in peripheral blood and ICOS expression in the uterus of CD19^−/−^, BIL-10^−/−^ and µMT dams. Our data show that B cell-specific IL-10-signaling is essential for a balanced maternal immune response to an inflammatory stimulant that cannot be hampered without IL-10-secreting B cells.

## 1. Introduction

Preterm birth (PTB), the delivery of an infant at less than 37 weeks’ gestational age, is still a major global obstetric problem. PTB is the leading cause of perinatal morbidity and mortality in developed countries [1,2]. PTB is regarded as a complex syndrome initiated by multiple mechanisms, including infection or inflammation and other immunologically mediated processes [3].

We have recently introduced B cells as central players in human pregnancy and of relevance for PTB [4,5]. B cells are mostly known as mediators of humoral immune responses since they are able to secrete antibodies following activation and differentiation. Despite this, B cells play multifaceted roles as enhancers and regulators of immunity. They act as professional antigen-presenting cells (APCs), provide co-stimulation and thereby contribute to autoimmunity and alloimmunity [6]. B cells can release pro-inflammatory cytokines and thereby further trigger inflammation and influence the differentiation of other immune cells such as T cells. B cells can also secrete anti-inflammatory cytokines. These so-called Regulatory B cells (Bregs) are activated by toll-like receptors (TLRs) rather than antigen receptors and operate earlier than regulatory T cells (Tregs) [7]. Breg cells suppress T effector cells, B effector cells, NK cells and monocytes/macrophages among others [8].

During pregnancy, a finely coordinated maternal immune balance is required for implantation, pregnancy maintenance and delivery. B cells contribute to the establishment of a tolerant environment by producing protective asymmetric antibodies and exerting immune-regulatory functions [9,10].

*Escherichia coli* is a gram-negative bacterium and the most common pathogen associated with bacteriuria, linked to urinary tract infections. *E. coli* is the most common bacterial infection in pregnancy, and also increases the risk of maternal and neonatal morbidity and mortality such as preeclampsia (PE), preterm birth (PTB), intrauterine growth restriction (IUGR), low birth weight and early-onset sepsis [11,12]. The endotoxin of gram-negative bacteria, lipopolysaccharide (LPS), is known to cause PTB in rodents and contributes to human PTB [3]. TLR4 plays a central role in the recognition of LPS, activates myeloid differentiation primary response protein 88 (MyD88)-dependent and MyD88-independent signaling pathways and induces the production of pro-inflammatory cytokines, which activate potent immune responses [13,14]. Besides, another TLR protein, RP105, also regulates B cell responses to LPS stimulation [15]. RP105 is complexed with MD-1 and mainly expressed on murine and human mature B cells. RP105/MD-1 is indispensable for TLR4/MD-2-dependent proliferation and the differentiation of marginal zone B cells into IgM-secreting plasma cells [16], while regulating B cell sensitivity to LPS [15]. Moreover, B cell stimulation with an anti-RP105 antibody and LPS induced CD19 phosphorylation, and CD19-deficient mice present an altered LPS response, both indicating an important role for RP105/CD19 interactions for activation of B cells [17].

We previously reported that the loss of B cell-specific MyD88 or IL-10 expression hinders an appropriate in utero development and influences the susceptibility to LPS-induced PTB [18]. Moreover, we found an enhanced uterine artery resistance in BIL-10^−/−^ dams at gestational day (gd)10 [18]. Here, we addressed the functional consequences of B cell-specific loss of CD19 or IL-10 expression and B cell deficiency for LPS-mediated PTB.

## 2. Materials and Methods

### 2.1. Animal Experiments and Sample Collection

Animal experiments complied with the ARRIVE guidelines and were carried out according to institutional guidelines after ministerial approval and in conformity with the European Communities Council Directive (EU Directive 2010/63/EU for animal experiments; approval number: 42502-2-1332 Uni MD). Eight–twelve weeks old C57BL/6 (H2^b^) wildtype littermates (WT), CD19^cre/wt^ IL-10^flox/flox^ B cell-specific IL-10 deficient mice (BIL-10^−/−^) (IL-10^flox/flox^ mice were provided by Axel Roers (TU Dresden, Dresden, Germany [19]; CD19^−/−^ mice lacking CD19 expression (B6.129P2(C)^Cd19tm1(cre)Cgn^/J CD19^−/−^ [20]; obtained from Jackson Laboratories, Bar Harbor, ME, USA) and B cell deficient µMT mice (B6.129S2-*Ighm^tm1Cgn^*/J [21]; obtained from Jackson Laboratories, Bar Harbor, ME, USA) were bred in the animal facility at the Medical University of Magdeburg. BALB/c (H2^d^) males were purchased from Janvier (Le Genest-Saint-Isle, France). Mice were kept under controlled light and humidity conditions in a 12 h light cycle. Chow and water were given ad libitum. Virgin female WT, BIL-10^−/−^, CD19^−/−^, µMT mice were mated with BALB/c males. Females were inspected twice a day for vaginal plug. The presence of a vaginal plug was designated as day 0 of pregnancy (gd0).

According to our previous study [18], 0.4 mg LPS (#L4391; Sigma Aldrich, Munich, Germany)/kg BW were i.p. injected on gd16 [18]. Mice were sacrificed 24 h later. Peritoneal lavage was taken by flushing the peritoneum with 1ml of 0.9% NaCl solution (Fresenius Kabi, Bad Homburg, Germany), centrifuged at 250× *g*, collected and stored at −80 °C. The cell pellet was resuspended in FACS buffer and stained (see below). Blood was obtained by puncture of the heart and stored in heparinized tubes on ice. Blood was centrifuged at 7000 rpm for 10 min. at RT. Serum was collected and stored at −80 °C.

### 2.2. Cell Staining and Flow Cytometry

Single-cell suspensions from peritoneal lavage, blood and uterus were stained with cell surface marker-directed antibodies and a viability dye for 30 min at 4 °C. For detection of the intracellular expression of cytokines and transcriptions factors, cell suspensions were fixed overnight using Fix and Perm (#00-5123-43, 00-5223-56; ebioscience, Dreieich, Germany), and stained with the intracellular antibodies in perm buffer (#00-8333-56) for 30 min at 4 °C. A list of the used antibodies is found in Appendix A while the composition of the panels is presented in Appendix A. Measurements were performed using an Attune NxT flow cytometer (Thermo Fisher Scientific, Dreieich, Germany). Data were analyzed with FlowJo software.

### 2.3. Cytokine and Chemokine Detection in Sera and Peritoneal Lavage

Cytokines in sera and peritoneal lavage were measured by the cytometric bead array (CBA) mouse Th1/Th2/Th17 Cytokine Kit from BD Biosciences (#560485), Heidelberg, Germany, according to the supplier’s recommendation. Briefly, cytokine capture beads were incubated with recombinant standards or samples and the PE-conjugated detection antibodies. The capture antibodies were directed against IL2, IL-4, IL-6, IFN-γ, TNF-α, IL-17A and IL-10. Chemokine levels (IP-10, LIX, MDC, MIP-1α, MIP-1β, Eotaxin, BLC, KC, MCP-1, MIP-3α, RANTES, TARC and MIG) were determined with Legendplex murine proinflammatory chemokine panel (13-plex; #740007) assay following supplier’s recommendation (Biolegend, San Diego, CA, USA). Briefly, standards or samples were incubated with premixed beads, followed by the detection antibodies and finally Streptavidin-PE.

### 2.4. Measurement of Progesterone Concentrations

The levels of progesterone in serum and peritoneal lavage were determined according to the manufacturer’s recommendation (#582601; Cayman, Ann Arbor, MI, USA). Briefly, diluted serum samples and undiluted peritoneal lavage were incubated with progesterone AChE tracer and progesterone ELISA antiserum, developed with Ellman’s reagent and read on an ELISA reader (BioTek Synergy HT, Agilent Technologies, Winooski, VT, USA).

### 2.5. Data Analysis and Statistics

Statistical analysis was performed using GraphPad Prism 8.0 software. Normality of distribution was determined by a Shapiro–Wilk test. Data were analyzed by either One-way ANOVA, followed by Dunnett’s multiple comparisons test or a Kruskal–Wallis test, followed by Dunn’s multiple comparisons test or a Mann–Whitney U test. Significance was defined as follows: * *p* < 0.05, ** *p* < 0.01, *** *p* < 0.001, **** *p* < 0.0001.

## 3. Results

### 3.1. Loss of B Cell-Specific IL-10 Influences Maternal Well-Being after LPS-Induced Preterm Birth

Under the experimental conditions that we used, PTB occurs about 18h following an LPS injection [18]. 24 h after LPS, WT dams did not deliver and presented a healthy appearance. CD19^−/−^ and µMT B cell-deficient mice delivered preterm, recovered from early birth and showed only low burden (scoring: Appendix A). However, BIL-10^−/−^ dams appeared to be severely affected by delivery (Figure 1). They presented signs of lethargy, tachypnea, and tremor and were cold when they were touched. Signs of maternal sickness were e.g., the maternal liver, which looked grey and bloodless, the gut, which appeared to be collapsed, and the bloody peritoneal cavity (PC). One out of six dams died in the course of PTB. This clearly indicates that B cell-specific IL-10 deficiency impacts maternal health during preterm delivery.

### 3.2. B Cell-Specific IL-10 Deficiency Impacts on the Cellular Composition and Cytokine Concentrations in Peritoneal Cavity after LPS Challenge at gd16

The peritoneal cavity (PC) is the primary site of inflammation in our model of LPS-induced PTB (17), and the pregnant uterus is located within. Therefore, we measured the cellular distribution in control and LPS-treated dams. As expected, compared to WT dams, µMT mice lacked B cells in PC (Appendix A), but had enhanced frequencies of CD8+ T cells (Appendix A), CD4+ T cells (Figure 2A) and CD11c+ dendritic cells (DC) (Figure 2B), which decrease after LPS challenge in all mouse strains except in BIL-10^−/−^ dams (Figure 2A,B). LPS treatment was associated with an increased percentage of Ly6G+CD11b- granulocytes in all mothers, which was significantly higher in BIL-10^−/−^ compared to WT dams (Figure 2C). The macrophages from WT and CD19^−/−^ dams upregulated NOS2 after LPS, while BIL-10^−/−^ and µMT mice failed to do so (Figure 2D).

In the peritoneal lavage, LPS-treated BIL-10^−/−^ dams had a higher level of IL-6 (Figure 3A) and TNF-α (Figure 3B), but no detectable IL-10 compared to WT dams (Figure 3C).

These data imply that the percentages and function of peritoneal immune cells are changed in BIL-10^−/−^ dams upon LPS challenge compared to the other groups. Some alterations overlap with observations in B cell-deficient µMT mice, but some are uniquely connected to the inability of B cells to secrete IL-10.

### 3.3. CD19 Deficiency or B Cell Specific IL-10 Knockout Influence the Level of Cytokines and Chemokines Observed in Blood from Pregnant Animals That Received LPS

Among all investigated mouse strains, the serum levels of IL-6 and TNF-α were enhanced following LPS (Appendix A). IL-17A was lower in the serum of LPS-treated WT dams compared to BIL-10^−/−^ mice (Figure 4A). The level of several chemokines increased after LPS in all mouse strains, among them IP-10, MIP-1α, MIP-1β and eotaxin (Table 1).

However, the serum level of RANTES was not significantly changed in WT dams upon LPS treatment, but increased in CD19^−/−^, BIL-10^−/−^ and µMT mice 24 h after LPS (Figure 4B). MCP-1 was lower in WT dams compared to CD19^−/−^ mice after LPS challenge (Figure 4C). The progesterone levels remained unchanged in WT LPS dams but decreased in LPS-treated CD19^−/−^ mice and significantly in LPS-challenged BIL-10^−/−^ and µMT dams (Figure 4D).

Treatment of dams with LPS at gd16 did not affect the frequency of B220+ B cells in the blood of WT mice, but a reduced percentage of B cells was found in CD19^−/−^ and BIL-10^−/−^ mice (Appendix A). In the spleen and paraaortic lymph nodes (PLN), the percentage of B cells was not altered by the LPS challenge (Table 2 and Table 3). In WT, there were also no changes in the frequencies of IL-10-expressing B cells, IL-10+ CD4+ and IL-10+ CD8+ T cells (Figure 4E–G) following LPS, but LPS-treated CD19^−/−^ and BIL-10^−/−^ mice had lower IL-10+B220+ frequencies (Figure 4E). The percentages of IL-10+CD4+ and IL-10+ CD8+ T cells were lower in CD19^−/−^, BIL-10^−/−^ and µMT mice, independent of treatment, compared to LPS WT dams (Figure 4F,G). The frequency of IL-10+CD8+ T cells was higher in LPS-treated WT than BIL-10^−/−^ dams in the spleen and PLN and there was a similar trend also in IL-10+ CD4+ T cells and IL-10+CD4+CD25+Foxp3+ Treg, but this did not reach statistical significance (Table 2 and Table 3).

Peripheral B cells may travel to lymph organs and act there as APCs. Therefore, we determined the expression of CD80, CD86, PD-L1, PD-L2 and MHC class II by B cells 24 h after the LPS challenge. However, none of these molecules was altered by LPS treatment in B cells, either in B cell-deficient mice and B cell-specific CD19- or IL-10 deficient mice, or in WT controls (data not shown). Despite this, we determined that the LPS challenge induced the expression of PD-1 by CD4+ T cells (Figure 4H) and CD8+ T cells (Figure 4I) in WT dams. The frequency of PD-1+CD4+ T cells was higher in LPS-treated WT than in CD19^−/−^ PBS and LPS-treated µMT mice (Figure 4H), while the frequency of PD-1+CD8+ T cells was enhanced in WT LPS but no modifications were observed in all other investigated groups. We also investigated the expression of costimulatory molecules by CD4+ and CD8+ T cells in the spleen and PLN. The expression of ICOS by CD4+ T cells was higher in LPS-treated WT than BIL-10^−/−^ dams in the spleen and PLN (Table 2 and Table 3). Additionally, in PLN, LPS-challenged WT dams showed increased percentages of ICOS+CD8+ T cells and PD-1-expressing CD4+ T cells (Table 2 and Table 3).

The findings suggest that WT dams present a moderate induction of inflammatory cytokines and chemokines and only full B cell-competent WT dams have B cells and T cells secreting IL-10 in sufficient concentrations to protect against LPS-induced PTB. The data also indicate that IL-10-secreting B cells are needed to induce IL-10-secreting T cells.

### 3.4. B Cell-Specific IL-10 Deficiency Influences the Frequency of Uterine Cell Populations

In the uterus, the frequency of CD11c+ cells was reduced in LPS-treated BIL-10^−/−^ mice compared to LPS WT dams (Figure 5A). Despite the fact that uterine frequency of B220+ B cells was not changed by pregnancy or LPS treatment (Figure 5B), alterations in B cell subpopulations were present: LPS challenge upregulated the expression of TACI by B cells in WT dams, which was higher than in LPS-treated BIL-10^−/−^ mice (Figure 5C). The frequency of CXCR4+ B cells decreased 24 h after LPS in WT dams and was also lower in LPS-treated BIL-10^−/−^ mice compared to WT dams (Figure 5D). While no changes were found for the percentages of uterine CD4+ and CD8+ T cells (data not shown), the frequency of CD4+CD25+Foxp3+ Treg cells was diminished in LPS-treated BIL-10^−/−^ mice compared to LPS WT dams (Figure 5E). BIL-10^−/−^ mice had lower percentages of ICOS-expressing CD4+ T cells (Figure 5F) as well as CD8+ T cells (Figure 5G) and no detectable IL-10+ B cells in the uterus (Figure 5H). Compared to LPS-treated WT mice, BIL-10^−/−^ dams had more uterine TNF-α+ B cells (Figure 5I). WT CD4+ T cells induced IFN-γ 24 h after LPS, which was enhanced compared to all other groups (Figure 5J).

These observations refer to an imbalance between regulatory and inflammatory immune cells in the pregnant uterus of BIL-10^−/−^ mice upon an LPS challenge to inflammatory cells with fatal consequences.

## 4. Discussion

In the last couple of years, B cells have emerged as important players in pregnancy. We and others showed that B cells contribute to an immune balance in early pregnancy via multiple mechanisms of action [22,23,24,25,26]. Besides maturating to plasma cells and secreting antibodies [27], they secrete soluble factors that modulate T cell function, most prominently IL-10. IL-10 secretion by B cells was shown to be essential if pregnancy is jeopardized by inflammation as in the case of infections [28]. In mice, we showed that µMT mice lacking mature B cells, mice lacking CD19 expression (CD19^−/−^) or B cell-specific IL-10 deficient mice (BIL-10^−/−^) not only present smaller fetuses but also deliver preterm in response to LPS doses that do not induce PTB in WT controls [18]. Within the present paper, we extend the knowledge on how the absence of IL-10 secreting B cells impacts on cellular immune balance during pregnancy and LPS-induced murine PTB. This is analyzed together with the absence of mature B cells (µMT mice) or CD19 expression by B cells to better understand the different contributions of the missing B cell subpopulations.

It was shown before that IL-10-deficient (null mutant) mice have apparently normal pregnancies but are more susceptible to preterm fetal loss and display more inflammation in the serum and gestational tissues (uterus and placenta) [29]. However, the source of IL-10 during pregnancy was not addressed. Here, we confirm that B cell-specific IL-10 deficiency is related to increased susceptibility to LPS-induced PTB. Furthermore, the LPS challenge impacts maternal health during preterm delivery as well. In contrast to WT mice that did not deliver upon LPS challenge at the concentrations used here, CD19^−/−^, µMT and BIL-10^−/−^ mice delivered after LPS application at gd16, therefore, earlier. While CD19^−/−^ and µMT dams were only slightly affected in their general condition, LPS-treated BIL-10^−/−^ dams showed moderate burden and sickness. This was associated with increased IL-6 and TNF-α and decreased IL-10 levels in the peritoneal lavage and increased IL-17A and RANTES concentrations in the maternal serum. Increased RANTES and MCP-1 levels were also detected in the serum from CD19^−/−^ and µMT mice. We previously observed enhanced inflammatory cytokine concentrations in human PTB [4,5,30]. Others found this connection also in murine PTB models [31,32]. Several hematopoietic and non-hematopoietic cells expressed RANTES (regulated upon activation, normal T-cell expressed and secreted)/CCL5 (CC chemokine ligand 5). The chemokine acts as a potent chemoattractant for many cell types including monocytes, NK cells, memory T cells, granulocytes and dendritic cells [33]. The expression of RANTES was shown to dramatically increase at inflammatory sites [34]. Increased RANTES levels were associated with spontaneous PTB within 14 days after presentation with symptoms of preterm labor with [35]. In addition, RANTES was increased in the maternal plasma and placenta in preeclampsia patients [36]. In a murine LPS-induced PTB model, MCP-1 was determined in the maternal serum and uterus, in the myometrium and endometrium, indicating that MCP-1 potentially contributes to the pro-inflammatory immune response that leads to PTB [37]. Our data indicate that B cell-specific IL-10 contributes to resistance to preterm birth by downregulating inflammatory cytokines and chemokines in the serum and peritoneum with RANTES, IL-17 and MCP-1 as potential targets.

In our study, the absence of mature B cells, CD19+ B cells and B cell-specific IL-10 deficiency altered the cell frequencies in the peritoneum with increased numbers of CD4+ T cells and CD11c+ DCs as well as insufficient upregulation of NOS2 by macrophages after LPS. 24 h after LPS treatment, higher percentages of granulocytes and lower numbers of macrophages were found in the peritoneal cavity. The percentages of granulocytes were higher in LPS-treated BIL-10^−/−^ mice compared to LPS-treated controls. It is well established that human and murine parturition is associated with an influx of granulocytes in gestational tissues [38,39,40]. This is more pronounced in PTB [41]. Inflammatory immune responses alter the survival and differentiation of macrophages, based on the microenvironment. M1 macrophages are characterized by the production of NO through induction of NOS2 synthesis, helping to clear the inflammation [42,43]. µMT and BIL-10^−/−^ dams failed to induce NOS2+ macrophages after the LPS challenge. Induction of NOS2 by peritoneal macrophages was dependent on MyD88 and TLR4 expression in murine PTB [44]. Besides, reduced expression of NOS2 by alveolar macrophages was found in preterm human neonates with fulminant early-onset pneumonia, a disorder that is associated with ascending intrauterine infection [45].

Healthy pregnancies are associated with increased IL-10 levels [46,47]. IL-10-deficient mice and µMT B cell-deficient mice presented an increased susceptibility to LPS-induced preterm delivery compared to WT mice [29,48]. In our study, we found that the absence of CD19-expressing B cells, B cell specific IL-10 deficiency or mature B cells was associated with decreased IL-10 and PD-1 expression by CD4+ and CD8+ T cells in the blood. The inhibitory receptor PD-1 and its ligand PD-L1 help to maintain peripheral tolerance by inhibiting the activation of self-reactive lymphocytes. Blockage of PD-1 signaling abrogates the protective effect of regulatory T cells in pregnancy [49]. PD-1 further induces apoptosis of paternal antigen-specific T cells during pregnancy [50] and the interplay PD-1/PD-L1 is important for Treg cell development in a healthy pregnancy [51]. PD-1 expression by murine T cells in a healthy pregnancy was associated with enhanced IL-10 expression [52]. PD-1 and PD-L1 expression is also important for the polarization of decidual macrophages to an anti-inflammatory M2 phenotype in an early human pregnancy [53].

It was reported that the number of CD11c+ DCs increases in the uterus from gd5 on and reaches a plateau at gd17, indicating an immune-regulatory function throughout gestation at the feto-maternal interface [54,55]. In the uterus of LPS-treated BIL-10^−/−^ mice, we detected decreased numbers of CD11c+ DCs compared to WT dams. It was shown recently that uterine CD11c+ cells induce the development of paternal antigen-specific Treg cells [56]. Therefore, it is not surprising that BIL-10^−/−^ mice also had lower uterine CD4+CD25+Foxp3+ Treg cells when compared to WT dams. This seems to be a characteristic of the genotype of the mice rather than the LPS challenge since the treatment did not influence the frequency of CD11c+ DCs or Treg cells. It also goes in line with our previous observation that B cell-derived IL-10 was able to support the conversion of naive T cells into Treg cells [28].

We observed an increased percentage of B cells expressing the transmembrane activator and CAML interactor (TACI) in WT mice upon an LPS challenge. TACI expression was previously reported in human term placentas [57]. TACI is an immune regulatory molecule that promotes the differentiation and survival of plasma cells [58]. Since plasma cells are a known source of IL-10 [59], enhanced TACI expression in WT dams may contribute to their enhanced IL-10 expression. Besides, expression of TACI by B1b cells was shown to be induced by TLR signaling and provided a heightened protective response [60]. CD19^−/−^ and BIL-10^−/−^ dams failed to induce TACI. Patients with defective CD19 alleles presented a defective upregulation of TACI following TLR9 stimulation [61]. It was further shown that the expression of TACI was greater in IL-10-expressing B cells than in IL-10-negative B cells [62], which properly contributes to the lower frequency of TACI+ B cells in BIL-10^−/−^ dams.

Further, we observed a lower CXCR4 expression by uterine B cells in BIL-10^−/−^ mice compared to control WT B cells. Since CXCR4 is involved in B cell maturation, trafficking and humoral immunity [63], these functions may be affected in BIL-10^−/−^ mice. The percentage of CXCR4-expressing B cells was decreased in WT mice and unaltered in CD19^−/−^ mice after an LPS challenge. It was reported that LPS induced CXCR4 expression by peritoneal B1 cells, which then actively migrated out of the peritoneal cavity [64]. Whether such a pathway also exists in uterine CXCR4+ B cells remains to be investigated.

Although we found an altered PD-1 expression dependent on the mouse strain in the blood, the PD1-PD-L1/L2 pathway was not affected in the uterus. However, we investigated other costimulatory molecules in the uterus and found in BIL-10^−/−^ dams reduced frequencies of ICOS-expressing CD4+ and CD8+ T cells compared to WT mice. The ICOS/ICOS-L costimulatory pathway is important for the maintenance of feto-maternal tolerance since ICOS+ T cells were shown to secrete IL-10 [65]. Blockade of this pathway increased fetal resorption [66], in the same way as we have previously seen in BIL-10^−/−^ mice [18]. This was associated with enhanced CD8+ T effector cells and reduced CD8+ Treg cells [66]. A reduced number of ICOS-expressing peripheral T cells were determined in patients with spontaneous preterm labor, preeclampsia and HELLP syndrome [67,68]. Since we have shown that patients with spontaneous preterm labor, preeclampsia or HELLP syndrome had elevated levels of pro-inflammatory cytokines such as IL-6 and a diminished percentage of IL-10-secreting B cells [4,5,30], ICOS-expressing T cells may be involved in the regulation of IL-10-secreting B cells.

Both costimulatory molecules, PD-1 and ICOS, are also expressed in peripheral, uterine and placental T follicular helper cells (Tfh). Tfh cells are pivotal to efficient humoral immunity and important for a successful pregnancy [69,70,71].

Taken together, our data suggest that B cell-competent WT mice have the ability to induce an immune response to eliminate the inflammatory stimulus and immunologically tolerate and support the unborn at once. The absence of IL-10-producing B cells has a deep impact on several immune cell types, such as macrophages, dendritic cells and T cells. Consequently, an inflammatory insult such as LPS induces a break in feto-maternal tolerance resulting in preterm delivery, characterized by an overwhelming release of inflammatory mediators and a defective upregulation of protective signaling molecules such as PD-1 or ICOS, and finally also affects maternal well-being.

## Figures and Tables

**Figure 1 cells-11-02690-f001:**
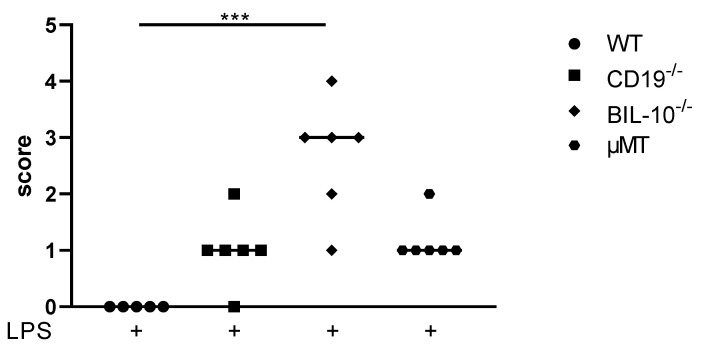
Scoring of maternal well-being 24 h after LPS injection at gd16. The score rating to define maternal well-being 24 h after LPS injection at gd16 was dependent on the measures and observation of mice and is shown for WT, CD19^−/−^, BIL-10^−/−^ and µMT mice. Data were analyzed by the Kruskal–Wallis test, followed by Dunn´s multiple comparisons test. *N* = 5–6 mice/group; *** *p* < 0.001.

**Figure 2 cells-11-02690-f002:**
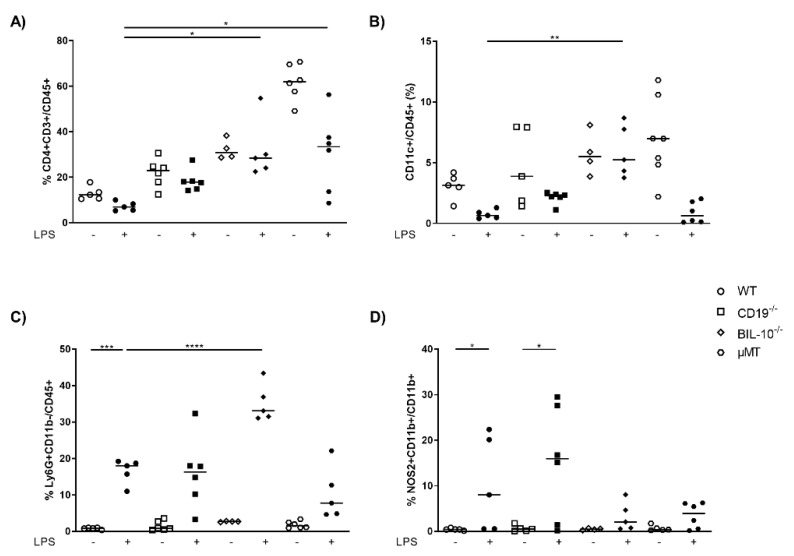
Loss of B cell-specific IL-10 influences the cellular composition in peritoneal cavity after LPS challenge at gd16. 24 h after PBS or LPS injection in WT, CD19^−/−^, BIL-10^−/−^ and µMT mice, percentages of CD4+ T cells (**A**), CD11c+ DC (**B**) and Ly6G+CD11b- granulocytes (**C**), analyzed within the CD45+ cell population, the frequency of NOS2+CD11b+ (**D**) analyzed within the CD11b+ cells, were determined in PC by flow cytometry. Data were analyzed by Kruskal–Wallis test, followed by Dunn’s multiple comparisons test. *N* = 4–6 mice/group. * *p* < 0.05, ** *p* < 0.01, *** *p* < 0.001, **** *p* < 0.0001.

**Figure 3 cells-11-02690-f003:**
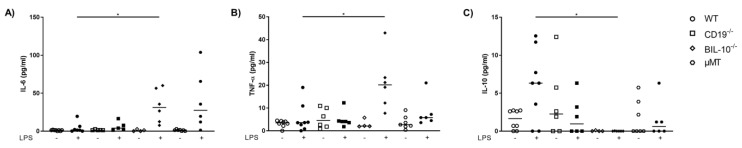
Loss of B cell-specific IL-10 influences the cytokine release into the peritoneal cavity after LPS challenge at gd16. In the peritoneal lavage, IL-6 (**A**), TNF-α (**B**) and IL-10 (**C**) were measured 24 h after PBS or LPS injection in WT, CD19^−/−^, BIL-10^−/−^ and µMT mice. Data were analyzed by Kruskal–Wallis test, followed by Dunn’s multiple comparisons test. *N* = 4–6 mice/group. * *p* < 0.05.

**Figure 4 cells-11-02690-f004:**
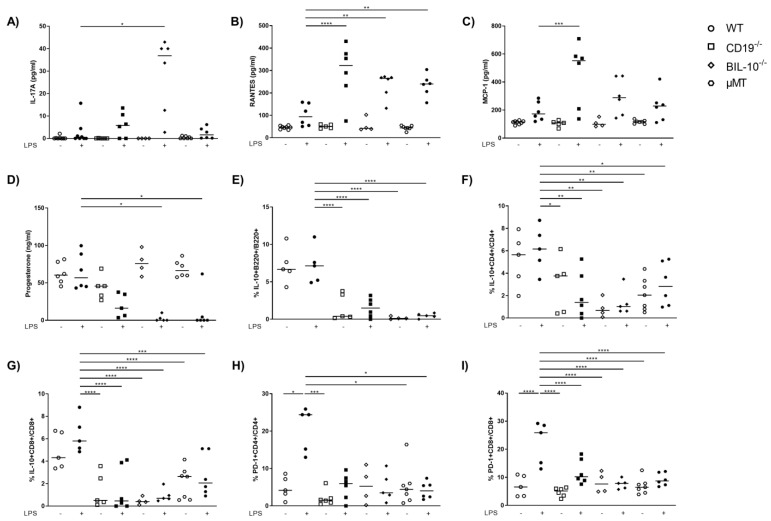
Alterations in B cell-specific CD19 or IL-10 expression and B cell deficiency influence the level of cytokines and chemokines in the blood. The level of IL-17A (**A**), RANTES (**B**) and MCP-1 (**C**) in serum of WT, CD19^−/−^, BIL-10^−/−^ and µMT mice was measured by flow cytometry, the progesterone level was determined by ELISA (**D**). The frequencies of IL-10+B220+ B cells (**E**), IL-10+CD4+ (**F**), IL-10+CD8+ T cells (**G**) and the expression of PD-1 by CD4+ (**H**) and CD8+ T cells (**I**) were determined by flow cytometry. Data were analyzed by Kruskal–Wallis test, followed by Dunn´s multiple comparisons test. N = 4–6 mice/group. * *p* < 0.05, ** *p* < 0.01, *** *p* < 0.001, **** *p* < 0.0001.

**Figure 5 cells-11-02690-f005:**
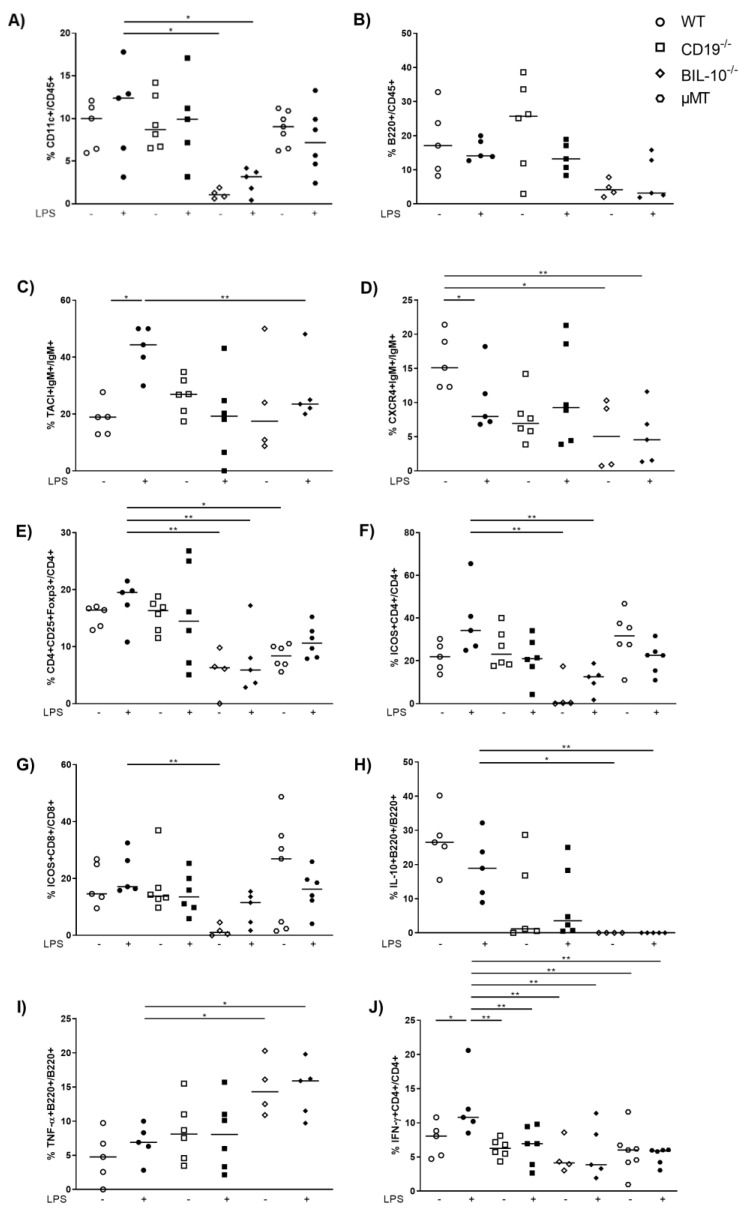
B cell-specific IL-10 deficiency influences the uterine cell populations. In the uterus, the frequencies of CD11c+ cells (**A**), B220+ B cells (**B**), TACI+IgM+ cells (**C**), CXCR4+IgM+ B cells (**D**), CD4+CD25+Foxp3+ Treg cells (**E**), ICOS+CD4+ T cells (**F**), ICOS+CD8+ T cells (**G**), IL-10+ B220+ B cells (**H**), TNF-α+ B220+ (**I**) and IFN-γ+CD4+ T cells (**J**) were detected by flow cytometry. Data were analyzed by Kruskal–Wallis test, followed by Dunn´s multiple comparisons test. *N* = 4–6 mice/group. * *p* < 0.05, ** *p* < 0.01.

**Table 1 cells-11-02690-t001:** Chemokine levels (in pg/mL) in serum of PBS- and LPS-treated WT, CD19^−/−^, BIL-10^−/−^ and µMT mice. Chemokines were measured by flow cytometry. Presented are the mean values; the *p* values were calculated by One-way ANOVA or Kruskal–Wallis test after Shapiro–Wilk test.

Chemokine	WT PBS	WT LPS	CD19^−/−^ PBS	CD19^−/−^ LPS	BIL-10^−/−^ PBS	BIL-10^−/−^ LPS	µMT PBS	µMT LPS	*p*
IP-10	13.9	32.6	13.6	45.4	19.0	28.3	7.7	15.6	<0.0001
LIX	334.0	587.0	733.5	841.5	205.5	341.3	662.7	440.7	0.0176
MDC	86.8	150.2	121.8	136.5	108.7	106.0	88.2	93.3	0.0386
MIP-1α	28.6	72.2	39.9	97.6	41.1	85.7	31.7	59.4	<0.0001
MIP-1β	26.2	124.9	17.0	100.3	36.5	76.0	9.0	38.6	0.0002
Eotaxin	179.6	350.6	200.3	278.3	199.1	237.4	144.8	202.7	0.0057
KC	130.5	195.7	134.8	314.9	142.9	185.9	112.8	141.3	<0.0001
MCP-1	111.6	190.2	103.7	457.0	108.1	294.9	116.2	228.8	<0.0001
MIP-3α	82.2	100.2	79.5	86.8	72.6	88.0	63.8	68.9	0.0268
RANTES	46.1	101.2	50.6	292.6	55.7	234.3	44.5	234.0	<0.0001
TARC	42.7	102.2	52.0	94.8	49.1	74.3	39.2	44.3	0.0180
MIG	38.3	115.3	55.5	117.1	58.3	99.9	40.2	69.6	<0.0001

**Table 2 cells-11-02690-t002:** Immune cell populations in spleen. Shown are the the mean frequencies of B220+ B cells, ICOS-expressing CD4+ and CD8+ T cells, PD-1-expressing CD4+ and CD8+ T cells as well as IL-10-expressing CD4+, CD4+CD25+Foxp3+ Treg cells and CD8+ T cells in spleen, determined by flow cytometry. *N* = 4–6 mice/group, the *p* values were analyzed by One-way ANOVA or Kruskal–Wallis test after Shapiro–Wilk test.

Cell Population	WT PBS	WT LPS	CD19^−/−^ PBS	CD19^−/−^ LPS	BIL-10^−/−^ PBS	BIL-10^−/−^ LPS	µMT PBS	µMT LPS	*p*
B220+	61.0	57.3	42.7	42.9	54.5	40.6			0.2972
CD4+CD25+Foxp3+	8.4	8.4	7.0	9.7	3.7	4.9	4.7	4.2	0.0019
ICOS+CD4+	8.6	5.6	5.2	5.1	1.6	2.0	3.3	2.6	<0.0001
ICOS+CD8+	2.0	2.3	2.7	2.5	0.8	0.6	1.4	1.5	0.0461
PD-1+CD4+	4.9	3.4	5.2	3.1	1.3	0.9	2.5	1.8	0.0036
PD-1+CD8+	2.0	3.1	2.6	3.7	1.0	1.0	2.7	2.4	0.0352
IL-10+CD4+	5.2	5.0	1.9	1.9	1.0	0.9	2.3	2.5	0.0036
IL-10+CD4+ Treg	6.5	7.0	3.5	3.4	1.2	2.1	3.8	3.4	0.0555
IL-10+CD8+	5.6	4.9	2.7	2.3	0.6	1.3	2.6	2.4	0.0055

**Table 3 cells-11-02690-t003:** Immune cell populations in paraaortic lymph nodes (PLN). Shown are the the mean frequencies of B220+ B cells, ICOS-expressing CD4+ and CD8+ T cells, PD-1-expressing CD4+ and CD8+ T cells as well as IL-10-expressing CD4+, CD4+CD25+Foxp3+ Treg cells and CD8+ T cells in PLN, determined by flow cytometry. *N* = 4–6 mice/group; the *p* values were analyzed by One-way ANOVA or Kruskal–Wallis test after Shapiro–Wilk test.

Cell Population	WT PBS	WT LPS	CD19^−/−^ PBS	CD19^−/−^ LPS	BIL-10^−/−^ PBS	BIL-10^−/−^ LPS	µMT PBS	µMT LPS	*p*
B220+	65.0	62.0	46.3	56.1	46.5	61.9			0.0041
CD4+CD25+Foxp3+	8.3	10.9	6.2	9.3	2.4	2.8	4.4	6.9	0.0002
ICOS+CD4+	12.9	17.1	9.8	9.8	0.7	3.0	7.6	7.4	<0.0001
ICOS+CD8+	14.1	14.1	7.2	12.2	0.6	1.1	9.8	8.5	0.0062
PD-1+CD4+	12.9	17.1	9.8	9.8	1.59	2.8	7.6	7.4	<0.0001
PD-1+CD8+	9.8	9.0	9.8	9.7	1.3	1.9	7.0	9.1	0.1422
IL-10+CD4+	1.9	4.82	1.8	1.3	0.3	0.7	1.9	2.2	0.0967
IL-10+CD4+ Treg	2.2	2.5	3.5	2.5	0.3	0.9	2.6	2.0	0.1539
IL-10+CD8+	3.5	3.3	2.7	1.6	0.4	0.7	2.4	2.7	0.0037

## Data Availability

The datasets in this study are available from the corresponding author upon reasonable request.

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
