# Peer review of "IL-10 Producing B Cells Protect against LPS-Induced Murine Preterm Birth by Promoting PD1- and ICOS-Expressing T Cells"

_cells, 2022, doi:10.3390/cells11172690_

Round 1

Reviewer 1 Report

In this manuscript, the authors demonstrate that IL-10-secreting B cells could interfere the preterm birth. Furthermore, the authors go on to show that BIL-10-/- mice upregulate inflammatory cytokines and downregulate the anti-inflammatory mediators. The authors suggested that fewer Tregs cells and downregulation of PD1 and ICOS in B cell-deficient mice resulting an increase in PTB. The manuscript is well written; however, some clarifications would be useful.

Manuscript Concerns:

Method section, please improve below to help readers easily read out:

1.     Please label the catalog number for all the materials.

2.     Is there any lectures or source for CD19-/-, BIL-10-/-, and µMT mice?

3.     Briefly describe the protocol you used for cytokine and chemokine analysis, especially indicate the maker you detected in different batches.

4.     Why you use the Kruskal-Wallis test instead of ANOVA?

Results section:

1.     In figure S1A, why the percentage of B220+ was higher in the BTL-10-/- group compared with the WT group? As an important immune response organ, did you measure these cell components at SP or Lymph node?

2.     In figure 2, it is difficult to understand the label of the cellular composition, such as “% CD4+CD3+/CD45+”, NOS+CD11b+/CD11b+”, did this mean you calculate the ratio of CD4+CD3+ in CD45+ cells? Or just frequencies of CD45+CD4+CD3+ cells? The data showed that compared with the WT+LPS group, the CD4+ population in the BIL-10-/- -LPS and µMT-LPS group are significant increase. The comparison should be made between WT-LPS and deficient mice-LPS, not sure if it is meaningful because the baseline in transgenic mice had differences. Several figures had a comparison between the WT+LPS group and deficient mice-LPS, I would suggest getting rid of them. Comparisons between WT-LPS and deficiency mice-LPS are meaningful.

3.     As you indicated that “the percentage of CD11b+Ly6G- macrophages decreased in all investigated strains (Figure 2D)”, please address the comparison. Moreover, the statistics didn’t show significant changes when we compared the +LPS group and -LPS group.  Please check all the descriptions and it is necessary to claim your group comparison.

4.     In Table 1, what’s the unit of each measurement, what’s the comparison of p-value. You mentioned the ANOVA test here and missed it in the method section. SD value or SEM should be added here.

5.     In figure 4G, the percentage of CD8+IL-10+ in CD19-/--LPS group and BIL-10-/--LPS group is also very low compared with WT-LPS, did you check the IL-10 positive cells in different cell types in these two mice model? Such as CD4+, CD8+ T cell, and Treg cells. Otherwise, it is difficult to say whether the protection against the effect of IL-10 responds by B cells or T cells.

6.     For PD1 and ICOS, Spleen and lymph nodes should be get involved in the research as they are the critical source for T cells and B cells response in the body.

Author Response

We thank the Reviewer for the time and the dedication to review our manuscript. We highly appreciate the suggestions and the constructive criticisms of reviewer #1 that helped us to further improve the manuscript. Please find attached a point-by-point reply to reviewer #1 comments, changes are marked in red in the revised manuscript file.

Manuscript Concerns:

  1. We added the catalogue numbers for all used materials in this study.
  2. We added references and sources for CD19-/-, IL-10flox/flox and µMT mice.
  3. We added a brief protocol for cytokine and chemokine analysis. We bought the kits at once, therefore no batch differences were detected.
  4. In dependence of the Shapiro-Wilk normality test, we used either One-way ANOVA (all data passed the normality test) or Kruskal-Wallis test (not all data passed the normality test).

Results section:

  1. We detected no statistical significant difference between WT and BIL-10-/- We measured the percentage of B220+ also in spleen and paraaortic lymph nodes. We added the results in Table 2 (spleen) and Table 3 (PLN).
  2. The percentages of the major cell populations, such as CD4+CD3+ T cells or CD11b+Ly6G+ cells, were always calculated within the CD45+ cells. However, the frequency of NOS-expressing CD11b cells were calculated within the population of CD11b+ cells (which were before calculated within the CD45+ cells). We clarified this in the Figure legend. We deleted all comparisons between WT LPS and deficiency mice-PBS.
  3. We deleted the Figure.
  4. We added the unit (pg/ml) and the ANOVA test in the method section. We did not add SD values for clarity of the table. However, we are happy to provide you with this information (see attached file)
  5. We measured the percentage of IL-10-expressing CD4+, CD4+CD25+Foxp3+ Treg cells and CD8+ T cells additionally in spleen and paraaortic lymph nodes. We added the results in Table 2 (spleen) and Table 3 (PLN).
  6. We measured the percentage of IL-10-expressing CD4+, CD4+CD25+Foxp3+ Treg cells and CD8+ T cells additionally in spleen and paraaortic lymph nodes. We added the results in Table 2 (spleen) and Table 3 (PLN).

Reviewer 2 Report

Dear authors,

thank you for your excelent research. The data presented in the article entitled „IL-10 producing B cells protect against LPS-induced murine 2 preterm birth by promoting PD1- and ICOS-expressing T cells” are the result of a more extensive research which reflect your interest on discovering immunological mechanims involved in the pathology of pregnancy complications. The article is well written and the conclusions extend your previous data.

You aimed to analyze the ability of IL-10 secreting cells to protect from PTB and the underlying mechanisms. Based on high interesting experimental research the article bring new knowledge about how the absence of IL-10 secreting B cells impacts on cellular immune balance along pregnancy and LPS-induced murine preterm birth. The research is conducted also in mice with absence of B cells (uMT mice), CD19 expression and wild type mice (controls) to better understand the contribution of each missing B cell subpopulation in these mechanisms.

In conclusion, you suggest that B cell-competent wild type mice have the ability to limit the immune response, to suppress the inflammatory stimulus and to assure an immunologically tolerance millieu for the fetus. Also, the absence of IL-10-producing B cells has a deep impact on several immune cell types, such as macrophages, dendritic cells and T cells. Consequently, an inflammatory insult likewise LPS induces a break of feto-maternal tolerance resulting in preterm delivery, characterized by an excessive release of inflammatory mediators, defective upregulation of protective signaling molecules such as PD-1 or ICOS and finally also affected maternal well-being.

Your results are in line with other observations which show that spontaneous preterm labour is often associated with an increased inflammatory response which is initiated before clinical symptoms appear (1,2).

I find that the results of the article are highly interesting and I recommend publication without changes. I am sure that the article will attract the attention of the readers and open the door for further studies.

Kind regards,

1. Keskin U, Ulubay M, Kurt YG, Fidan U, KoçyiÄŸit YK, Honca T, Aydin FN, Ergün A. Increased neopterin level and chitotriosidase activity in pregnant women with threatened preterm labor. J Matern Fetal Neonatal Med. 2015 Jun;28(9):1077-81. doi: 10.3109/14767058.2014.943174. Epub 2014 Jul 28. PMID: 25005858.

2. Navolan DB, Vladareanu S, Lahdou I, Ciohat I, Kleist C, Grigoras D, Vladareanu R, Terness P, Sas I. Early pregnancy serum neopterin concentrations predict spontaneous preterm birth in asymptomatic pregnant women. J Perinat Med. 2016 Jul 1;44(5):517-22. doi: 10.1515/jpm-2015-0081. PMID: 25918916.

Author Response

We thank the Reviewer for the time and the dedication to review our manuscript. We are very happy and honored after reading the comments. Thanks.

Round 2

Reviewer 1 Report

Agree to accept.